# Genomic and Transcriptional Profiling Analysis and Insights into Rhodomyrtone Yield in *Rhodomyrtus tomentosa* (Aiton) Hassk

**DOI:** 10.3390/plants12173156

**Published:** 2023-09-01

**Authors:** Alisa Nakkaew, Thipphanet Masjon, Supayang Piyawan Voravuthikunchai

**Affiliations:** 1Center for Genomic and Bioinformatics Research, Faculty of Science, Prince of Songkla University, Hat Yai 90110, Songkhla, Thailand; bsc0209.psu@gmail.com; 2Division of Biological Science, Molecular Biotechnology and Bioinformatics, Faculty of Science, Prince of Songkla University, Hat Yai 90110, Songkhla, Thailand; 3Center of Antimicrobial Biomaterial Innovation-Southeast Asia, Faculty of Science, Prince of Songkla University, Hat Yai 90110, Songkhla, Thailand; supayang.v@psu.ac.th

**Keywords:** *Rhodomyrtus tomentosa*, *matK*, ITS, transcriptional profile, rhodomyrtone, ZnSO_4_ stress, zinc transporter protein

## Abstract

*Rhodomyrtus tomentosa* is a source of a novel antibiotic, rhodomyrtone. Because of the increasing industrial demand for this compound, germplasm with a high rhodomyrtone content is the key to sustainable future cultivation. In this study, rhodomyrtone genotypes were verified using the plastid genomic region marker *matK* and nuclear ribosomal internal transcribed spacer ITS. These two DNA barcodes proved to be useful tools for identifying different rhodomyrtone contents via the SNP haplotypes C569T and A561G, respectively. The results were correlated with rhodomyrtone content determined via HPLC. Subsequently, *R. tomentosa* samples with high- and low-rhodomyrtone genotypes were collected for de novo transcriptome and gene expression analyses. A total of 83,402 unigenes were classified into 25 KOG classifications, and 74,102 annotated unigenes were obtained. Analysis of differential gene expression between samples or groups using DESeq2 revealed highly expressed levels related to rhodomyrtone content in two genotypes. semiquantitative RT-PCR further revealed that the high rhodomyrtone content in these two genotypes correlated with expression of zinc transporter protein (*RtZnT*). In addition, we found that expression of *RtZnT* resulted in increased sensitivity of *R. tomentosa* under ZnSO_4_ stress. The findings provide useful information for selection of cultivation sites to achieve high rhodomyrtone yields in *R. tomentosa*.

## 1. Introduction

*Rhodomyrtus tomentosa—*has long been used as a traditional medicine in Asian countries such as China, Vietnam, Indonesia, Malaysia, and Thailand. It has been reported that many parts of the plant contain various phytochemical compositions. Indigenous people in Southeast Asia use the berries as a remedy for dysentery and diarrhea. Parts of the roots and stem are used for stomach ailments and as a traditional medicine for women after childbirth. In Thailand, *R. tomentosa* is used as an antipyretic, antidiarrheal, and antidysenteric agent [1].

Rhodomyrtone, an acyl phloroglucinol compound from *R. tomentosa*, has a strong antibacterial activity against Gram-positive bacteria, including important pathogens such as *Enterococcus faecalis*, *Staphylococcus aureus*, *Staphylococcus epidermidis*, *Streptococcus gordonii*, *Streptococcus mutans*, *Streptococcus pneumoniae*, *Streptococcus pyogenes*, *Streptococcus salivarius*, and *Propionibacterium acnes*. In addition, it has notably specific activity against methicillin-resistant *S. aureus* (MRSA) with a minimum inhibitory concentration (MIC) and a minimum bactericidal concentration (MBC) of 0.39 to 0.78 mg/mL [2,3]. Its ability to prevent *Staphylococci* and *P. acnes* biofilm formation and kill mature biofilms has been reported and extensive data support the view that rhodomyrtone is a new candidate as a natural antibacterial drug [4,5,6].

*R. tomentosa* can produce an important substance for medical use and is currently being studied and used in clinical practice to treat antibiotic-resistant infections. In the cosmetic industry, it is used as an antibacterial agent against Gram-positive bacteria, etc. As a result, there is significant demand for rhodomyrtone from *R. tomentosa*, but there has been no cultivation of *R. tomentosa* for commercial use. In addition, the genetic information on *R. tomentosa* is limited. The genetic diversity of *R. tomentosa* has been studied in Malaysia and Hong Kong using inter-simple sequence repeat (ISSR) markers. The genetic diversity of 15 populations of *R. tomentosa* from Malaysia and 10 populations from Hong Kong showed that *R. tomentosa* has a relatively high level of genetic diversity and a low level of gene flow among *R. tomentosa* populations [7,8]. It is therefore necessary to promote conservation and increase cultivation to meet the increasing demand. Because of the value of rhodomyrtone, the germplasm of *R. tomentosa*, which produces a large quantity of this compound, should be selected for cultivation as a future industrial crop. Field evaluation could be optimized using molecular techniques such as marker-assisted selection which can improve breeding efficiency by allowing direct selection of a large set of rhodomyrtone genotypes. However, genotyping to identify the germplasm of *R. tomentosa* has not yet been reported. Marker-based breeding techniques have already been widely used for crop improvement and genetic conservation [9]. The use of molecular techniques has proven to be well-suited for the unambiguous identification of plant species because DNA molecules are found in all tissues and are not affected by the external or physiological conditions of the plant [10,11]. Nowadays, DNA barcoding and transcriptome is a combined technology used to find information about genotype and transcript profiles coordinately associated with the productivity of target plants [12,13]. Therefore, it would be interesting to explore whether genotype and transcript variants can be used as markers to help select and breed rhodomyrtone-producing elite germplasm. The Consortium for the Barcode of Life (CBOL) plant working group recommends two markers developed in 2009 from two plastid regions, namely rbcL and *matK* [14], which are used as DNA barcodes of terrestrial plants. In addition, some plant researchers suggest the use of ITS regions to support barcoding loci in order to increase the genotyping accuracy of this technique [15,16,17]. High-throughput sequencing using next-generation sequencing of the cDNA pool from which the transcriptomic information was obtained can also be used to obtain a large number of expressed sequences, which can in turn be used to design molecular markers associated with the coding genes and their function [18]. Furthermore, transcriptomics can provide important information about the evolution and conservation of genetic variation in plant species [19,20] and is a powerful strategy for linking genotype to phenotype; as such, it can be a successful tool for marker-based selection in medicinal plants [20,21]. Recently, transcriptomic analysis has been used to study specific genes for metabolite biosynthesis and expression patterns, which can then be used to determine synthesis and metabolic pathways. In addition, several reports have been published concerning the genomics, transcriptomics, metabolomics, and phenomics of certain plants such as *Sarcandra* sp. and *Dendrobium* sp., and these resources facilitate molecular breeding and gene mining [22,23,24,25,26]. 

However, there are no reports concerning the use of DNA barcoding for marker-assisted selection and transcriptome analysis in *R. tomentosa*. In this study, we collected wild *R. tomentosa* from fields in Surat Thani and Songkhla provinces in the Thai Peninsula. The genetic backgrounds of all the *R. tomentosa* samples were identified using DNA barcoding and classified into two groups based on high and low rhodomyrtone content using high-performance liquid chromatography (HPLC). In addition, to better understand rhodomyrtone-producing drivers at the genetic level, we performed a de novo transcriptome analysis to analyze gene expression. This allowed us to identify and observe the genes that were differentially expressed between high- and low-rhodomyrtone content samples. The results presented here provide useful information for future breeding programs.

## 2. Results and Discussion

### 2.1. Genotyping Analysis in Rhodomyrtus tomentosa

DNA barcoding has recently become a widely used and effective tool for rapid and accurate species discrimination. Several plastid DNA barcoding regions, e.g., *matK*, that function in all plant species have been evaluated and the combination of non-coding intergenic spacer identification (ITS) was found to be the best option for correct plant species discrimination [27,28]. In this work, based on the available information on plant DNA barcoding aimed at identifying the botanical origin of different species of *R. tomentosa*, different sites were selected in Surat Thani and Songkhla provinces in southern Thailand.

The total genomic DNA of the *R. tomentosa* samples was extracted and tested for quality using universal 18S rDNA primers (18SF/18SR), as described by Nakkaew et al. [29]. This showed positive amplification in all samples, confirming that there were no false negatives. Therefore, the loci *matK* and ITS were amplified from the *R. tomentosa* samples from Surat Thani (RtST1-RtST6) and Songkhla provinces (RtSK1-RtSK6), with direct sequencing of the two strands in opposite directions because the platform often does not allow for perfect resolution of the first 40–60 bp at the 5′ end of the sequence. The nucleotide sequences of *matK* and ITS were searched using BLASTN to identify homologous sequences. The nucleotide sequences were aligned using the Multiple Sequence Alignment option in ClustalX [30] and GeneDoc [31] standalone software, and the alleles amplified from the *R. tomentosa* samples were sequenced for further analysis. The alignment results showed that two single nucleotide polymorphisms (SNPs) existed in the sequences together with the *matK* and ITS loci (Figure 1A,B). The first SNP from the *matK* locus was located at the 569th nucleotide position and was found to be a transition SNP (C to T). The nucleotide ‘C’ was present in all the *R. tomentosa* samples from Surat Thani, namely, the Surat Thani genotype, whereas ‘T’ was present in all the samples from Songkhla province, namely, the Songkhla genotype. The second SNP, from the ITS1 locus, was a transitional SNP (from A to G) found at nucleotide position 561. The nucleotide ‘A’ was present only in the Surat Thani genotype, whereas nucleotide ‘G’ was present only in the Songkhla genotype. From the above results, it is evident that the two SNPs contribute to the differences in the genotyping of *R. tomentosa* from Surat Thani and Songkhla provinces. Similar results were found in relation to *Amomum villosum* Lour., which is used in traditional Chinese medicine. SNP genotypes were identified in the plastid genes rbcL., *matK*, and ITS regions which could be used as a multiregional DNA barcode to accurately identify Amomi Fructus landraces in different producing areas. This may imply the typing of a single nucleotide polymorphism based on DNA barcoding markers in order to identify the germplasm of Amomi Fructus [32]. In a recent study on the rhizomatous medicinal herb *Polygonatum odoratum*, plastid genomic information and nuclear SNP markers were used to assess the quality of *P. odoratum* cultivars and classify them as medicinal resources with premium medicinal and edible properties [33].

### 2.2. Rhodomyrtone Quantitation Analysis in Rhodomyrtus tomentosa

Based on genotyping analysis, *R. tomentosa* was divided into two groups: the Surat Thani and Songkhla genotypes. Next, the rhodomyrtone in these two genotypes was obtained via supercritical fluid extraction (SFE) and then determined using high-performance liquid chromatography (HPLC). The amounts of rhodomyrtone found in the two genotypes were very different. The average amounts of rhodomyrtone in the Surat Thani and Songkhla genotypes were 149.83 ± 0.08 and 25.89 ± 0.09 µg/mL, respectively (Figure 2). Based on the above results, in this study we describe the Surat Thani genotype (RtST) as a high-rhodomyrtone and the Songkhla genotype (RtSK) as a low-rhodomyrtone genotype.

### 2.3. De Novo Transcriptome Analysis of High- and Low-Rhodomyrtone R. tomentosa

Four cDNA libraries were prepared from the low- (RtSK1 and RtSK2) and high-rhodomyrtone genotypes (RtST1 and RtST2) of *R. tomentosa*. The libraries were sequenced using the BGISEQ-500 sequencer for transcriptome analysis. The libraries sequenced with paired-end sequence reads (PE) contained approximately 9.85, 9.97, 9.90, and 9.80 Gb of total clean bases from the RtSK1, RtSK2, RtST1, and RtST2 libraries, respectively (BioProject submission No. PRJNA956731). Then, the adapter and low-quality reads were trimmed, and the short reads (<20 bp) were removed. Quality analyses of the clean reads from the RtSK1, RtSK2, RtST1, and RtST2 libraries yielded approximately 86–92% at Q30. The raw clean reads of 65.68 Mb, 66.48 Mb, 66.03 Mb, and 65.34 Mb (Table 1), showing low- and high-rhodomyrtone genotyping, were subjected to assembly analysis using Trinity software [34]. The de novo assembly results had a total length of 81,844,035, 63,411,497, 58,750,215, and 81,846,852 bases, respectively, and were expressed in a total of 96,036 (157,501 transcript), 74,548 (118.921 transcript), 66,782 (103,157 transcript), and 98,941 (170,364 transcript) unigenes with an average length of the four libraries of 655 bp and an N50 length of 1155 bp (Table 1). In addition, the average GC content was 47.79% and the assembled transcripts that were longer than 500 bp were 37.55%, 40.53%, 43.83%, and 34.27% of each library. Then, all transcripts from the four libraries were collected for further analysis by clustering the assembled transcripts and using the TGI Clustering Tool (TGICL) [35] to remove redundant transcripts from each library and obtain all the unigenes and quality metrics (Table 2 and Figure 3). The length of most of the unigenes ranged from 300 bp to ≥3000 bp. In addition, 186,033 contigs were predicted with an average length of 584 bp, and 52,018 (30.08%) of the predicted contigs were longer than 500 bp. The high-quality sequencing results support the following functional annotations.

### 2.4. R. tomentosa Unigene Annotation and Classification

A total of 186,033 contig sequences were identified using the Nr database (Figure 4), the NCBI NT database, the Swiss Institute of Bioinformatics (Swiss-Prot) databases, the Kyoto Encyclopedia of Genes and Genomes (KEGG), the Clusters of Orthologous Groups of Proteins (COG) database, and the Gene Ontology (GO) resource with a statistical significance E value < 0.00001 (Table 3).

The 83,402 unigenes were analyzed for similarity to NR (nonredundant database) annotated plant datasets using BLAST searches with an e-value < 0.00001, and similarities with *Eucalyptus grandis* (53.49%), *Hordeum vulgare* subsp. Vulgare (4.32%), *Punica granatum* (1.06%), *Zea mays* (1.03%), and others (40.10%) were identified (Figure 5). These results are consistent as *E. grandis* is closely related to *R. tomentosa*.

In addition, the sequence homology of 83,402 unigenes was annotated. The results from the NT database showed that 58,242 unigenes matched the NCBI official protein database. Next, the unigenes were annotated by noting their alignment with the other functional databases: 54,573 (Swiss-Prot: 29.34%), 68,929 (KOG: 37.05%), 63,831 (KEGG: 34.31%), 48,439 (GO: 26.04%), and 74,102 (InterPro: 39.83%) unigenes were annotated; Figure 6 shows these annotations combined in a Venn diagram. In addition, for the functional annotation of transcription factors (TFs), 1933 unigenes were predicted based on the unigenes (54,672 Unigenes.cds) detected using Transdecoder [34]. Transcription factors (TFs) are important and regulate gene expression in various biological processes, including the secondary metabolism. The results showed that the TF family group were mainly enriched in the MYB and MYB-related genes (523 unigenes), followed by the C2H2 (155 unigenes), bHLH (126 unigenes), AP2-EREBP (121 unigenes), C3H (116 unigenes), NAC (115 unigenes), WRKY (108 unigenes), GRAS (79 unigenes), MADS (71 unigenes), G2-like (51 unigenes), mTERF(47 unigenes), Zn-clus (46 unigenes), and ABI3VP1 (45 unigenes) families (Figure 7). Recent reports identified that transcription factors contribute to the synthesis of bioactive compounds in medicinal plants. For example, the MYB, bHLH, and WD40 transcription factor families are important in flavonoid and anthocyanin biosynthesis [36,37], and it was found that the WRKY transcription factor family might be involved in the accumulation of kaempferol and quercitrin in *Camellia vietnamensis* [38]. 

A general function prediction was conducted for the annotated genes through referencing the 25 functional categories in the KOG database: such as 14,779 unigenes (14.20%) are general function prediction only; 13,003 unigenes (12.49%) involved in signal transduction mechanisms; 8767 unigenes (8.42%) in post-translational modification, protein turnover, and chaperones; 7084 unigenes (6.80%) in translation, ribosomal structure, and biogenesis; 5889 unigenes (5.66%) in carbohydrate transport and metabolism; 5434 unigenes (5.22%) with an unknown function; 4219 unigenes (4.05%) involved in defense mechanisms; 3418 unigenes (3.08%) in lipid transport and metabolism; 2513 unigenes (2.41%) in the biosynthesis, transport, and catabolism of secondary metabolites; 839 unigenes (0.81%) in nucleotide transport and metabolism, 628 unigenes (0.06%) in nuclear structure and the lowest, 67 unigenes (<0.06%) in cell motility (Figure 8).

The GO annotation identified 55 functional groups belonging to the three main GO ontologies: molecular function, cellular component, and biological process. Twenty-two groups, the majority, belonged to the domain of metabolic processes (33%, 20,500 unigenes), followed by cellular processes (32%, 20,116 unigenes), and biological regulatory processes (7%, 4131 unigenes) (Figure 9A).

There were 17 categories in the cellular component domain and these were mainly involved in cell functions (19%, 18,160 unigenes), cell parts (19%, 17,794 unigenes), and membrane (16%, 14,791 unigenes) (Figure 9B). Finally, 16 domains were found in the molecular function category. These were mainly involved in catalytic activity (43%, 25,711 unigenes), binding (40%, 24,001 unigenes), structural molecular activity (7%, 3898 unigenes), and the lowest with just 1 unigene was toxin activity (Figure 9C). These GO mapping results indicate that most DEGs responding to heat stress are involved in metabolic processes, cell and catalytic activity, and enzymatic activity. These affected activities indicate that heat stress treatment primarily causes functional changes in physiological metabolism and cell differentiation.

To identify active biological pathways in the transcriptome, sequences were mapped to the canonical reference pathways in the Kyoto Encyclopedia of Genes and Genomes (KEGG). A total of 63,831 sequences were classified using the KEGG database and grouped into five main categories. The highest results were found in metabolic pathways (64.15%, 41,974 unigenes), followed by genetic information processing pathways (24.18%, 15,824 unigenes). These were followed by the cellular processes and environmental information processing pathways with 4.63% and 4.59% (3028 and 3004 unigenes), respectively, and the lowest value was that for organismic systems (4.63%, 1602 unigenes) (Figure 10). Thus, the reported unigenes were found in the metabolic pathways subcategories such as biosynthesis of other secondary metabolites (2301 unigenes), metabolism of terpenoids and polyketides (1083 unigenes), and metabolism of other amino acids (1739 unigenes). These subcategories should therefore be considered as sequences that are potentially useful for defining metabolic pathways that might be related to the accumulation of rhodomyrtone in *R. tomentosa*.

### 2.5. Analysis of Differentially Expressed Genes of R. tomentosa

Based on the clean reads from four libraries with low (RtSK1 and RtSK2) and high rhodomyrtone contents (RtST1 and RtST2), an analysis of gene expression levels was performed. The gene expression levels in each library were calculated using RSEM and, to provide a more intuitive measurement, gene expressions at different FPKM intervals (FPKM ≤ 1, FPKM: 1~10, FPKM ≥ 10) were calculated for each library. In Figure 11, the depth of color indicates the different gene expression levels: FPKM ≤ 1 indicates an extremely low expression level and 1 < FPKM ≤ 10 indicates a high expression level. 

The gene expression levels of all 172,223 DEGs in the low- and high-rhodomyrtone *R. tomentosa* datasets were compared. The DEseq2 results were plotted using a Venn diagram, and scatter and volcano plots were used to show the expression levels of the different sample groups. From the scatter plot, it is clear that 4380 and 5513 DEGs were up- and downregulated, respectively, in the high-rhodomyrtone *R. tomentosa* datasets (Figure 12).

Additionally, a false discovery rate (FDR) < 0.05 and a |log2(fold-change)| > 1 were used as thresholds for identifying differentially expressed genes (DEGs) of the 9893 genes in the low- and high-rhodomyrtone samples of R. tomentosa. The DEGs were subjected to GO and KEGG enrichment analyses to determine the differences in biological processes and pathways between the low- and high-rhodomyrtone genotypes of R. tomentosa. Then, DEGs between the low- and high-rhodomyrtone genotypes were annotated using GO terms. In the biological processes category, the top GO terms were metabolic and cellular processes with 2083 and 1992 DEGs, respectively, followed by biological regulation, localization, and regulation of biological processes, which were assigned 378, 349, and 349 DEGs, respectively. In the cellular component category, there were only three terms, cellular anatomical unit, proteinaceous complex, and virion component, which were assigned 3086, 468, and 3 DEGs, respectively. In addition, the most common molecular function GO terms among the DEGs were catalytic activity, binding, and structural molecular activity, which were assigned to 2703, 2198, and 404 DEGs, respectively (Figure 13B).

Out of 134 KEGG pathways, all low- and high-rhodomyrtone DEGs were classified into seven branches of the KEGG functional classification and matched to five KEGG pathway branches, with 4463 associated with the metabolic branches. This was followed by genetic information processing, environmental information processing, cellular processes, and organismal systems branches, which were associated with 1751, 387, 253, and 201 DEGs, respectively (Figure 13B).

### 2.6. In Silico Analysis of Enhancement Gene Provides Insights into Rhodomyrtone Biosynthesis

Transcriptome profiling identified 9893 differentially expressed genes which were divided into the five KEGG pathway branches with the most abundant metabolites. The smallest P/Q values between the low- and high-rhodomyrtone genotypes were nucleocytoplasmic transport, RNA polymerase, the phenylpropanoid biosynthesis pathway, the plant MAPK signaling pathway, and plant hormone signal transduction (Figure 14A,B). Based on these results, analysis of differentially expressed unigenes between the two groups was performed to identify potential genes involved in regulating the rhodomyrtone content of *R. tomentosa*. Using the DEseq2 method [39], compared with the low-rhodomyrtone genotype, 4,380 genes were upregulated and 5513 genes were downregulated in the high-rhodomyrtone genotype. In addition, the fifteen genes with the highest and lowest expressions were identified in the high-rhodomyrtone groups. The results showed that several hormone-signaling transcription factors (CL14267.Contig2_All, CL12904.Contig1_All, CL11462.Contig1_All, and CL5679.Contig2_All) and stress signaling pathways (CL1945.Contig1_All and CL3612.Contig4_All) were related to rhodomyrtone in *R. tomentosa* (Table 4 and Figure 15) and were therefore identified as having possible involvement in rhodomyrtone biosynthesis. A candidate gene that may contribute to further improving yield, nutraceutical quality, and secondary metabolites in *R. tomentosa* is the zinc transporter protein (CL1945.Contig1_All), because zinc is essential for plant growth and is involved in the accumulation of phytochemicals [40,41].

### 2.7. Expression Analysis of Rhodomyrtone Related Genes through qRT-PCR Validation

To verify the accuracy of the RNA-seq results, three genes associated with rhodomyrtone were randomly selected for semiquantitative RT-PCR validation (CL11462.Contig1_All, CL1945.Contig1_All, and CL3457.Contig3 as shown in Appendix A) and 18S rRNA primers were used as internal controls. These checks amplified only the *Zn transporter*, CL1945.Contig1_All. The RNA-seq data and recent studies suggest that *Zn transporter* may be a candidate gene potentially related to rhodomyrtone biosynthesis. *Zn transporters* (*ZnTs*) are the proteins mainly involved in Zn transport within the plant and zinc plays a role in plant defenses against pathogens, including the enhancement of bioactive compounds in the plant cells [40,42]. Based on these results, *RtZnT* was selected for RT–qPCR validation to verify the reliability of the transcriptome data. Semiquantitative RT-PCR was performed to evaluate the *RtZnT* gene expression patterns. The total RNAs from the two low- and high- rhodomyrtone genotypes of *R. tomentosa* were amplified using *RtZnT*-specific primers and 18S rRNA primers as internal controls. *RtZnT* showed relatively strong transcription in the Surat Thani genotype (ST1-6) samples, whereas no *RtZnT* expression was detected in the Songkhla genotype (SK1-6) (Figure 16). These expression patterns were generally consistent with the RNA-seq results, confirming the accuracy of the transcriptome data.

In addition, the expression of *RtZnT* was observed in *R. tomentosa* grown hydroponically with two different ZnSO_4_ concentrations (100 mM and 500 mM) for 7 days. The genes were more highly expressed in the 500 mM ZnSO_4_ than in the 100 mM ZnSO_4_ conditions (Figure 17). This result is consistent with the effects observed in habanero pepper plants, in which ZnSO_4_ was found to increase the phenol and flavonoid content [43]. 

## 3. Materials and Methods

### 3.1. Plant Materials

Four biological groups of *Rhodomyrtus tomentosa* were collected from Banasan City, Surat Thani province (ST1 and ST2 from 8°54′22″ N 99°22′16″ E and 8°54′21″ N 99°22′18″ E, respectively) and Khlonghoykhong City, Songkhla province (SK1 and SK2 from 6°50′53″ N 100°22′31″ E and 6°50′54″ N 100°22′30″ E, respectively), Thailand, during the harvest season of May 2021. A quantity of 1 g of mature leaves from each group (high- and low-rhodomyrtone) was sampled with five replicates, frozen in liquid nitrogen, and then stored at −80 °C prior to de novo transcriptome analysis. In addition, 100 g of mature leaves from each group was sampled in triplicate and dried in a hot air oven at 60 °C for 16–18 h prior to rhodomyrtone extraction. 

### 3.2. DNA Barcoding Analysis of R. tomentosa Using matK and ITS Markers

DNA extraction and PCR amplification of the *matK* and ITS loci of *R. tomentosa*. Plant leaves were frozen in liquid nitrogen and ground into a fine powder. Genomic DNA was then isolated and purified using a Tissue Genomic DNA Mini Kit (Geneaid, New Taipei City, Taiwan) in accordance with the manufacturer’s instructions. The ribosomal region ITS was amplified with the primer pair ITSF (5′-CGTAACAAGGTTTCCGTAGGTGAAC-3′) and ITSR (5′-TTATTGATATGCTTAAACTCAGCGGG-3′ [44], and the chloroplastic *matK* loci were amplified with the primer pair MatKF (5′-CGTACAGTACTTTTGTGTTTACGAG-3′) and MatKR (5′-ACCCAGTCCATCTGGAAATCTTGGTTC-3′) [45]. The PCR reaction was performed in a total volume of 50 µL per 2 µL DNA template, 10 µL 5X HOT FIREPol Blend Master Mix (containing 10 Mm MgCl_2_), 2 µL of the mixed primer pair, and 36 µL deionized water. The PCR profile for *matK* and ITS was optimized via initial denaturation at 95 °C for 15 min, followed by 35 cycles starting with denaturation at 95 °C for 30 s and annealing at 60 °C for 30 s, followed by a final extension at 72 °C for 10 min. The amplified PCR products were sequenced in both directions using an automated DNA ABI PRISM 3700 sequencer, (Applied Biosystems, Waltham, MA, USA). Nucleotide sequence homology searches were performed using the BLASTN tool, and the alignment sequence was analyzed using the clustalX2.0 and GENDOC programs.

### 3.3. Extraction and Quantitative Determination of Rhodomyrtone 

A quantity of 10 g of *R. tomentosa* leaf powder obtained from Surat Thani and Songkhla provinces was filtered using a supercritical fluid extractor at the Office of Scientific Instrument and Testing (OSIT), Prince of Songkla University (PSU) (Hat Yai, Songkhla, Thailand) prior to extraction. The bioactive compound was extracted using 100% CO_2_ at a flow rate of 18 g/min, an extraction temperature of 55 °C, and a pressure of 250 bar. While one operating parameter was changed, the other operating parameters were kept constant. The obtained extract was dissolved in 100% DMSO and stored at −20 °C until further use. All samples were filtered through a 0.45 μm syringe filter. Aliquots of these samples were then taken for the determination of the total rhodomyrtone content at OSIT, PSU (Hat Yai, Songkhla, Thailand). Quantitative determination of rhodomyrtone substances was performed using an HPLC Agilent 1100 liquid chromatography system consisting of an Agilent Chemstation for GC, LC, LC /MSD, CE, UV-VIS detector (Agilent Technologies Inc., Santa Clara, CA, USA) and A/D Systems-Rev. 08. Ox. An aliquot of 20 uL of each sample solution was directly injected into the HPLC system using a mixture of acetonitrile (Merck Millipore, Burlington, MA, USA)) and water, which was run in the analytical Symmetry C8 column (4.6 × 150 mm, particle size 3 um) at 30 °C, and the peak was detected at 254 nm. The column was equilibrated for 6 min before each subsequent run and saturated with the mobile phase for at least 3 h before each assay. Rhodomyrtone (Sigma-Aldrich, St. Louis, MO, USA) was used as the internal standard for the assay.

### 3.4. De Novo Transcriptome Analysis of the High- and Low-Rhodomyrtone R. tomentosa

Five independent samples of leaves (*n* = 5) of equal weights were collected at the same stage for each of the low- (two replicate samples: SK1 and SK2) and high- (two replicate samples: ST1 and ST2) rhodomyrtone groups and sent to BGI-Shenzhen (Shenzhen, China) for library construction. Total RNA was extracted and measured using a NanoDrop 2000 spectrophotometer (Thermo Fisher Scientific Inc., Waltham, MA, USA) and RNA integrity was determined using an Agilent 2100 Bioanalyzer (Agilent Technologies Inc., Santa Clara, CA, USA). Sequencing libraries were constructed in accordance with the BGISEQ-500 library construction protocol. mRNA enrichment and reverse transcription to double-stranded cDNA (dscDNA) was conducted using N6 random primers and the synthesized cDNA was subjected to end repair and then 3′ adenylated. Adaptors were ligated to the ends of these 3′-adenylated cDNA fragments. The ligation products were purified and many rounds of PCR amplification were performed to enrich the purified cDNA template with PCR primers. Then, the PCR product was denatured using heat, and the single-stranded DNA was cyclized via splint oligo and DNA ligase and sequenced using the BGISEQ-500 platform (BGI, Shenzhen, China) to generate 100-bp paired-end reads. 

Then, quality control, de novo transcriptome composition, and functional annotation of genes were performed. A total of 4 RNA-seq datasets were used for the study, 2 of which were from a low-rhodomyrtone sample and 2 were from a high-rhodomyrtone sample deposited in the Sequence Read Archive database. The raw RNA-seq reads were trimmed and cleared of adapters using Trimmomatic v.0.36 [46] and AdapterRemoval (v. 2.1.3) [47]. Raw reads were then analyzed using FastQC [48] to calculate read quality metrics and Trinity was used to perform de novo assembly with clean reads. All unigenes were subjected to functional annotation analysis. Using Diamond, the assembled unigenes were aligned to the publicly available protein databases, including the NCBI nonredundant protein (NR), the Swiss-Prot protein (Swiss-Prot), Gene Ontology (GO), Clusters of Orthologous Groups (COG), and the Kyoto Encyclopedia of Genes and Genomes (KEGG). We used Blast2GO with a cutoff E value of 10^−5^ and InterProScan5 for InterPro annotation analysis. In addition, for CD prediction of unigenes we used Transdecoder (version 3.0.1) [34] and for plant TF prediction analysis we used getorf to find the ORF for each unigene, then aligned the ORF to TF domains using hmmsearch and identified TFs according to the PlantfDB database.

### 3.5. Unigene Expression Analysis

Unigene expression levels were calculated as Reads Per Kilobase Million Mapped Reads (RPKM), eliminating the effects of sequencing depth and gene length on gene expression levels and allowing direct data comparison with the DESeq2 method [39]. The expression levels of Unigenes involved in metabolic pathways related to seed oil accumulation were calculated. The DEGs between different time points were identified with padj < 0.05 and |log2 (fold change value)| ≥ 1.

### 3.6. Verification of Related Genes Using Semiquantitative RT-PCR 

The semiquantitative RT-PCR technique was used to test the reliability of the RNA-seq data. In this experiment, DEGs were randomly selected and primers were designed based on the RNA-seq contig data; information on the design of the semiquantitative RT-PCR primers is set out in Appendix A. Total RNA of *R. tomentosa* collected from different locations in Surat Thani and Songkhla provinces in southern Thailand was extracted using a Plant Total RNA Mini Kit (Geneaid, New Taipei City, Taiwan) according to the manufacturer’s protocol. The RT-PCR reactions contained 500 ng of RNA as template and were performed using the OneStep RT-PCR reaction kit in accordance with the manufacturer’s instructions (Qiagen, Hilden, Germany). The reaction was started at 50 °C for 30 min, followed by an initial PCR activation step at 95 °C for 15 min, then 40 cycles at 94 °C for 25 s, 60 °C for 25 s, and 72 °C for 25 s, and terminated by a 10 min incubation step at 72 °C. RT-PCR was performed to amplify a fragment of four DEG contigs and the 18S rRNA gene was amplified as a reference control. The RT-PCR products were separated using gel electrophoresis, visualized using ethidium bromide staining, and photographed to analyze expression levels using Quantity One software (Bio-Rad, Hercules, CA, USA).

## 4. Conclusions

This study is the first attempt to use the SNP haplotypes of *matK* and ITS barcodes to identify the rhodomyrtone genotype and determine the species composition of high-rhodomyrtone genotypes of *Rhodomyrtus tomentosa*. The results presented in this study could be a key to the sustainable cultivation of *R. tomentosa* plant genotypes with a high rhodomyrtone content. The results of RNA-Seq gene expression profiling showed that the two genotypes had different gene expression levels and that the highest gene expression level was related to the genotypes with a high rhodomyrtone content. The *zinc transporter protein* gene (CL1945.Contig1_All; *RtZnT*) was the most highly DEG in the Surat Thani genotype which had a high rhodomyrtone content, and the RNA-seq result was confirmed via semiquantitative RT-PCR analysis. Interestingly, we found that the expression of *RtZnT* was increased under ZnSO_4_ stress in *R. tomentosa*. These findings could be used in the production of *R. tomentosa* to improve the yields of rhodomyrtone.

## Figures and Tables

**Figure 1 plants-12-03156-f001:**
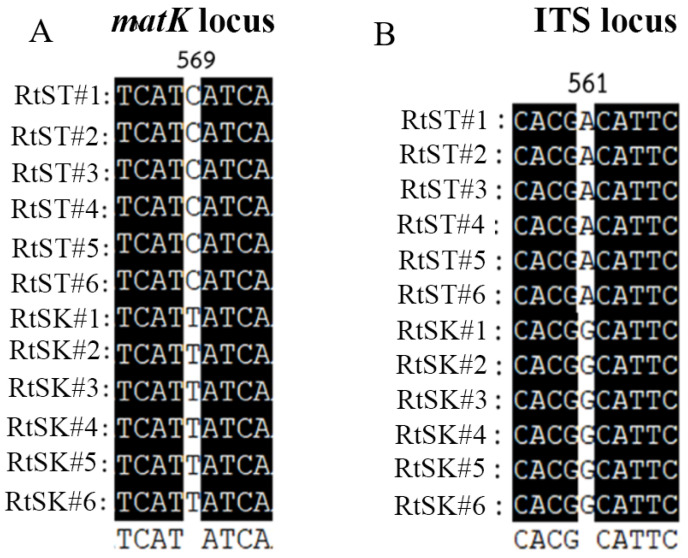
Alignment of the *matK* (**A**) and ITS loci (**B**) of the sequencing products of *Rhodomyrtus tomentosa* from Surat Thani (RtST1-RtST6) and Songkhla province (RtSK1-RtSK6) revealed the SNPs C569T and A561G, respectively.

**Figure 2 plants-12-03156-f002:**
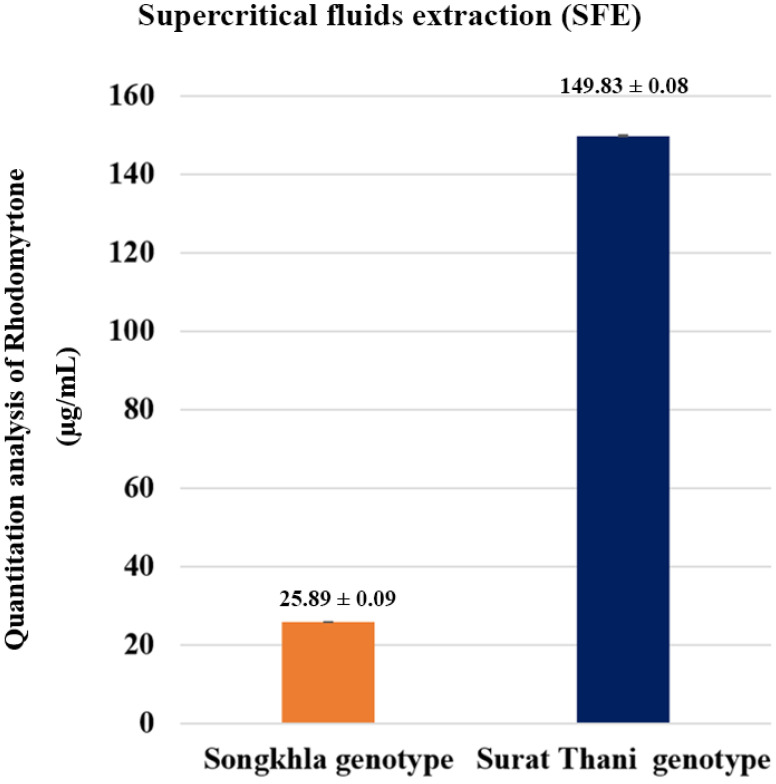
Quantitative analysis of rhodomyrtone-based SFE extracts from Surat Thani and Songkhla for genotyping of *R. tomentosa* using high-performance liquid chromatography (HPLC) at *p*-value < 0.01.

**Figure 3 plants-12-03156-f003:**
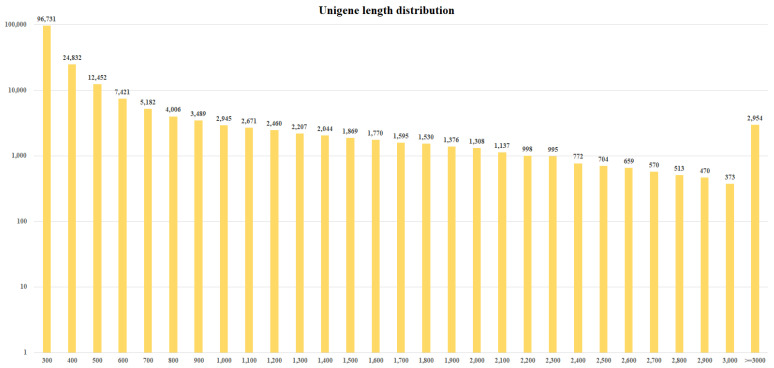
Unigene length distribution from the *R. tomentosa* transcriptome library.

**Figure 4 plants-12-03156-f004:**
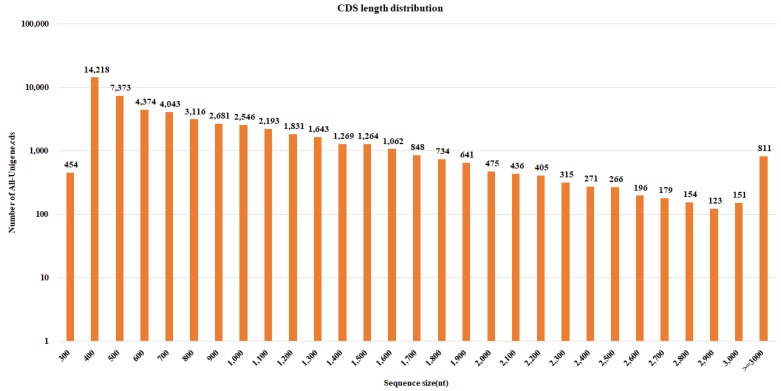
CDS length distribution (the X and Y axes represent the length and number of CDS, respectively).

**Figure 5 plants-12-03156-f005:**
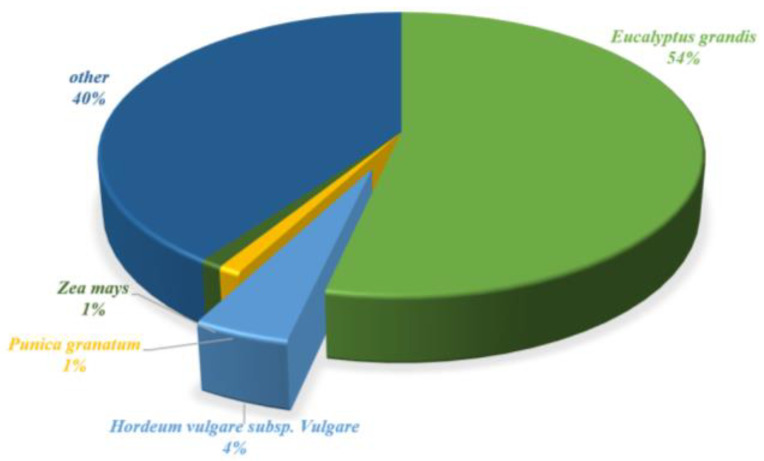
Distribution of NR annotated species from *R. tomentosa*.

**Figure 6 plants-12-03156-f006:**
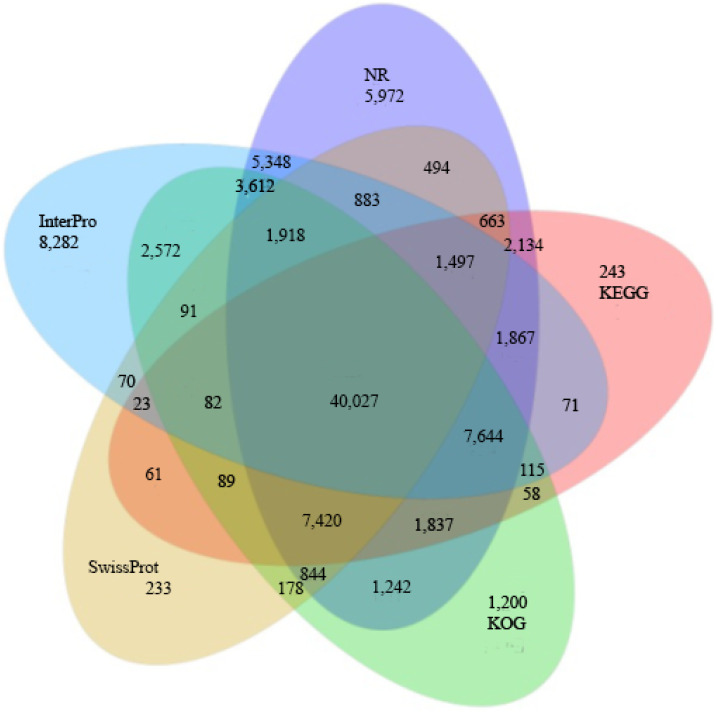
Venn diagram summarizing the *R. tomentosa* unigene annotations based on five databases NR (light yellow), KOG (light green), KEGG (light purple), SwissProt (light blue) and InterPpro (light blue). The number of unigenes with significant hits and the relationships between the databases are shown in each intersection and the results of the Venn diagram illustrating the different colors based on the relationships between the databases.

**Figure 7 plants-12-03156-f007:**
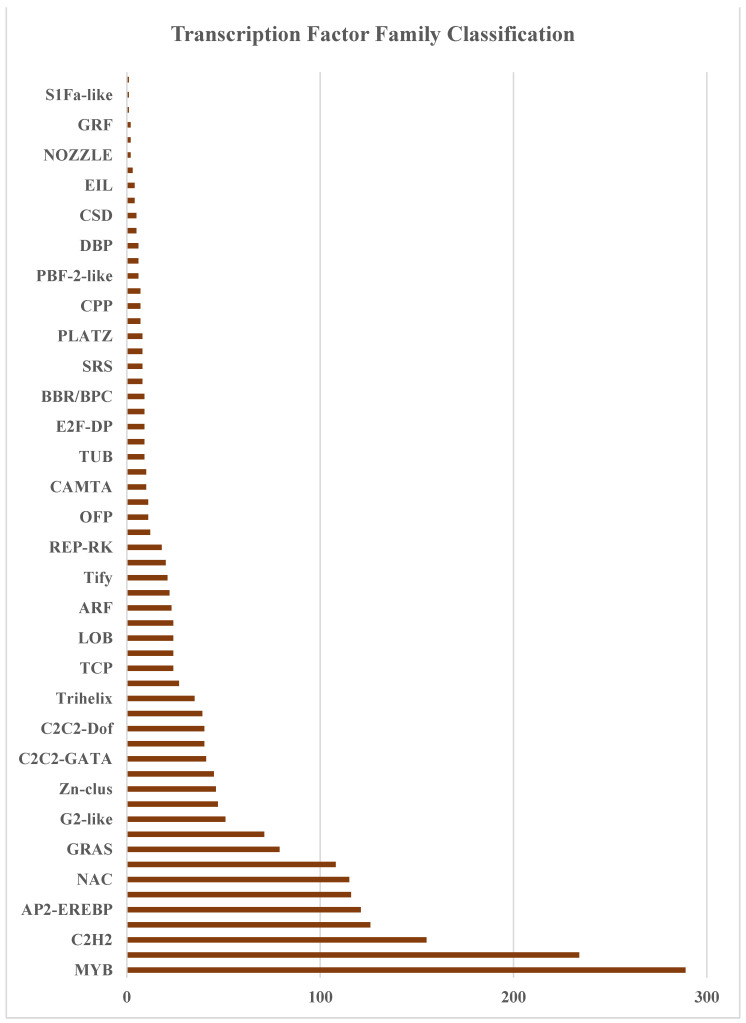
Unigene classification via transcription factor family.

**Figure 8 plants-12-03156-f008:**
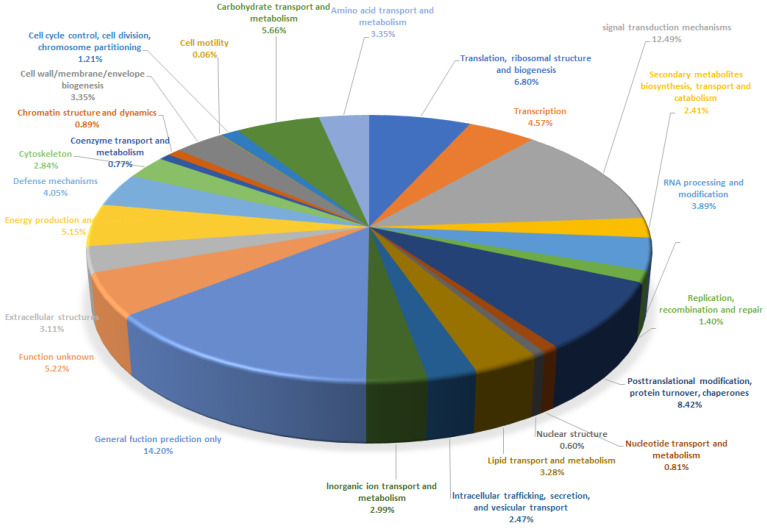
Functional distribution of KOG annotations.

**Figure 9 plants-12-03156-f009:**
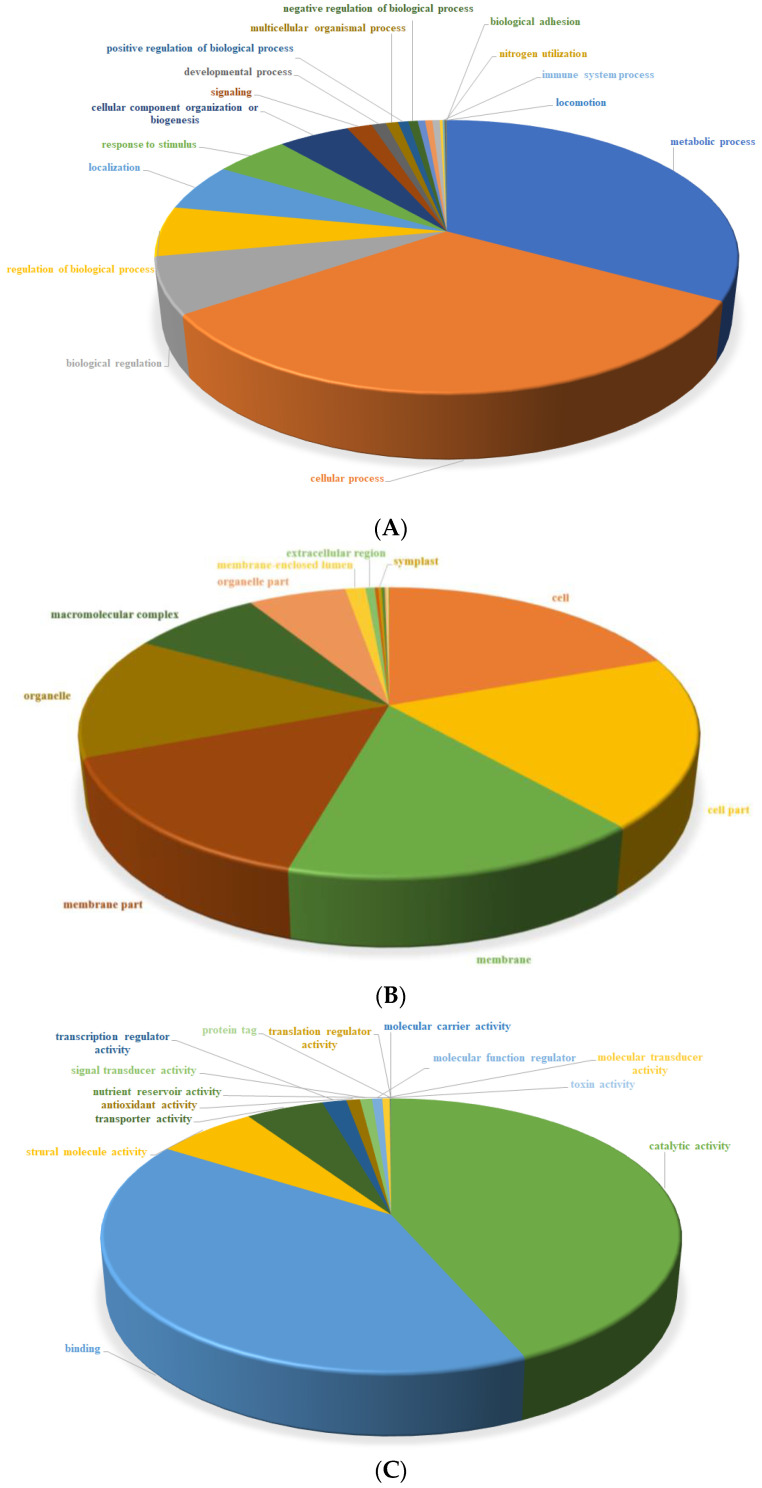
Functional distribution of GO annotation in all unigenes in with 3 Gene Ontology categories: biological process (**A**), cellular component (**B**), and molecular function (**C**).

**Figure 10 plants-12-03156-f010:**
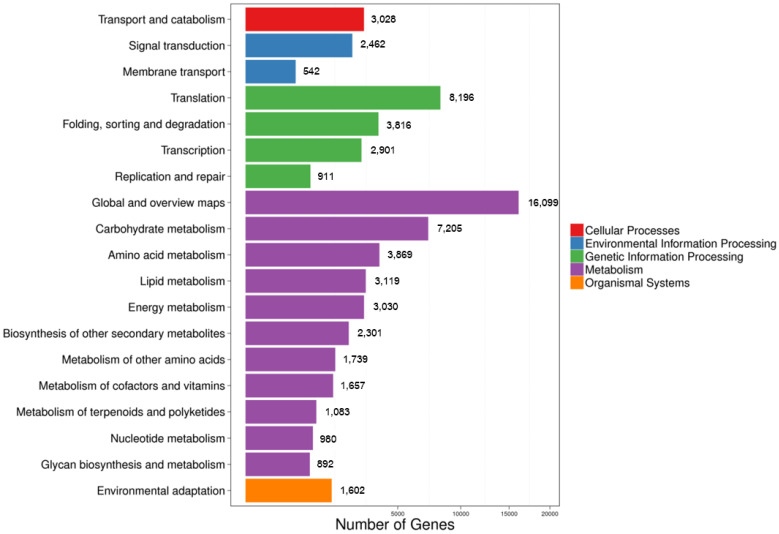
KEGG functional classification of *R. tomentosa* unigenes.

**Figure 11 plants-12-03156-f011:**
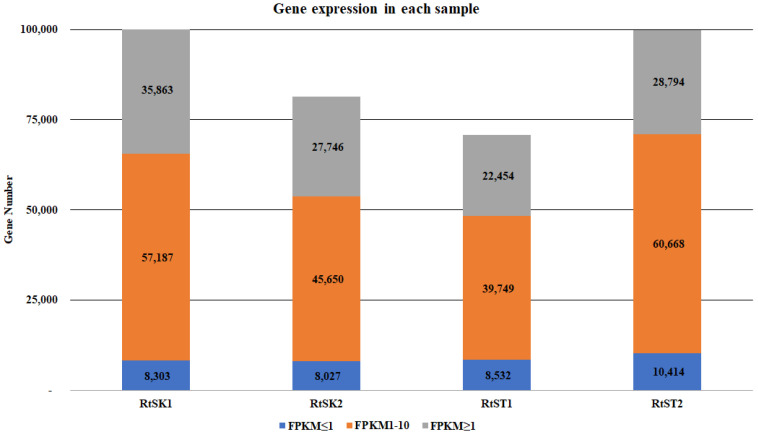
Gene expression distribution of the four low- (RtSK1 and RtSK2) and high-rhodomyrtone (RtST1 and RtST2) libraries of *R. tomentosa*.

**Figure 12 plants-12-03156-f012:**
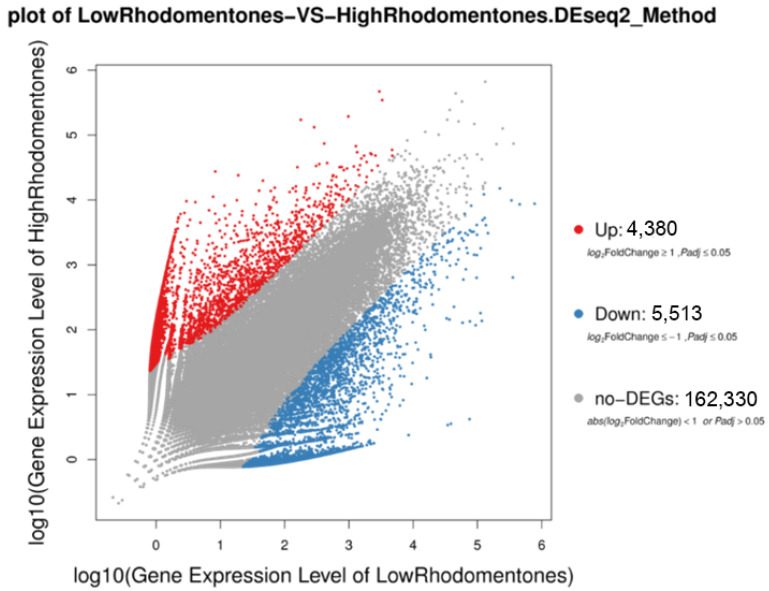
Scatter plot representing log10 transformed gene expression levels. The red color represents upregulated genes, the blue color represents downregulated genes, and the grey color represents nonsignificant differential genes.

**Figure 13 plants-12-03156-f013:**
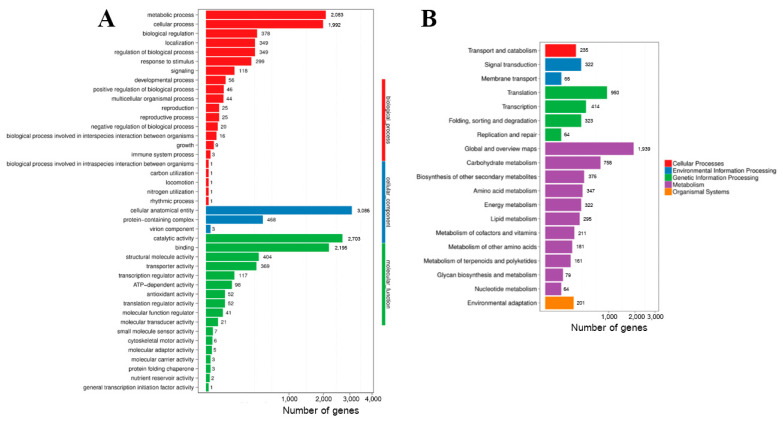
(**A**) GO classification and (**B**) pathway classification of DEGs. The x-axis represents the number of DEGs and the y-axis represents GO terms and KEGG functional classification.

**Figure 14 plants-12-03156-f014:**
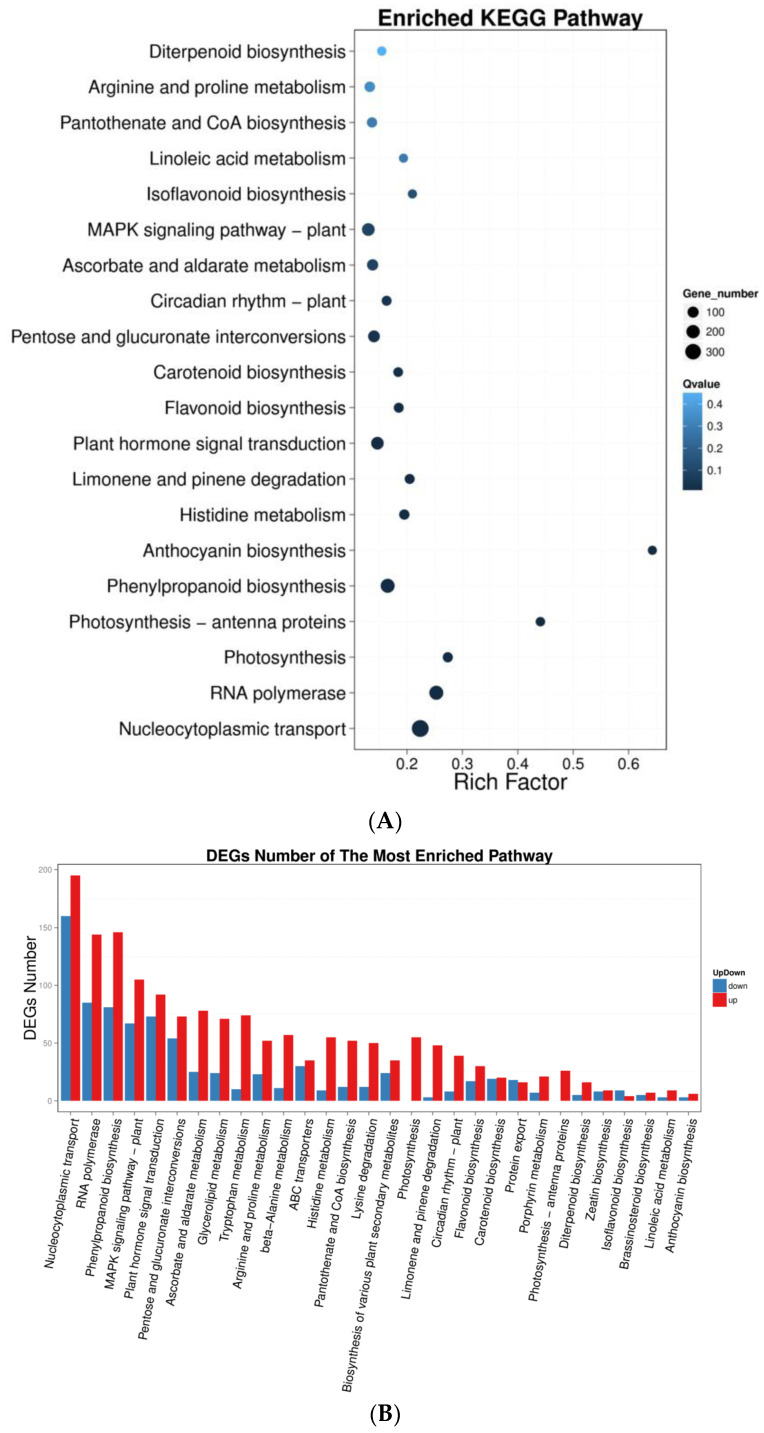
(**A**) Functional enrichment of DEGs by pathways showing the enrichment factor and pathway name. The color indicates the q value (high: white; low: blue), and the lower the q value, the more significant the enrichment. The dot size indicates the number of DEGs (the larger the dot, the larger the number). Rich factor refers to the enrichment factor value which is the quotient of foreground value (number of DEGs) and background value (total number of genes). The larger the value, the more significant the enrichment. (**B**) The result of functional enrichment of pathways for up- and downregulated genes. The x-axis represents the pathway terms and the y-axis represents the number of up- and downregulated genes detected in the high-rhodomyrtone genotypes using DEseq2.

**Figure 15 plants-12-03156-f015:**
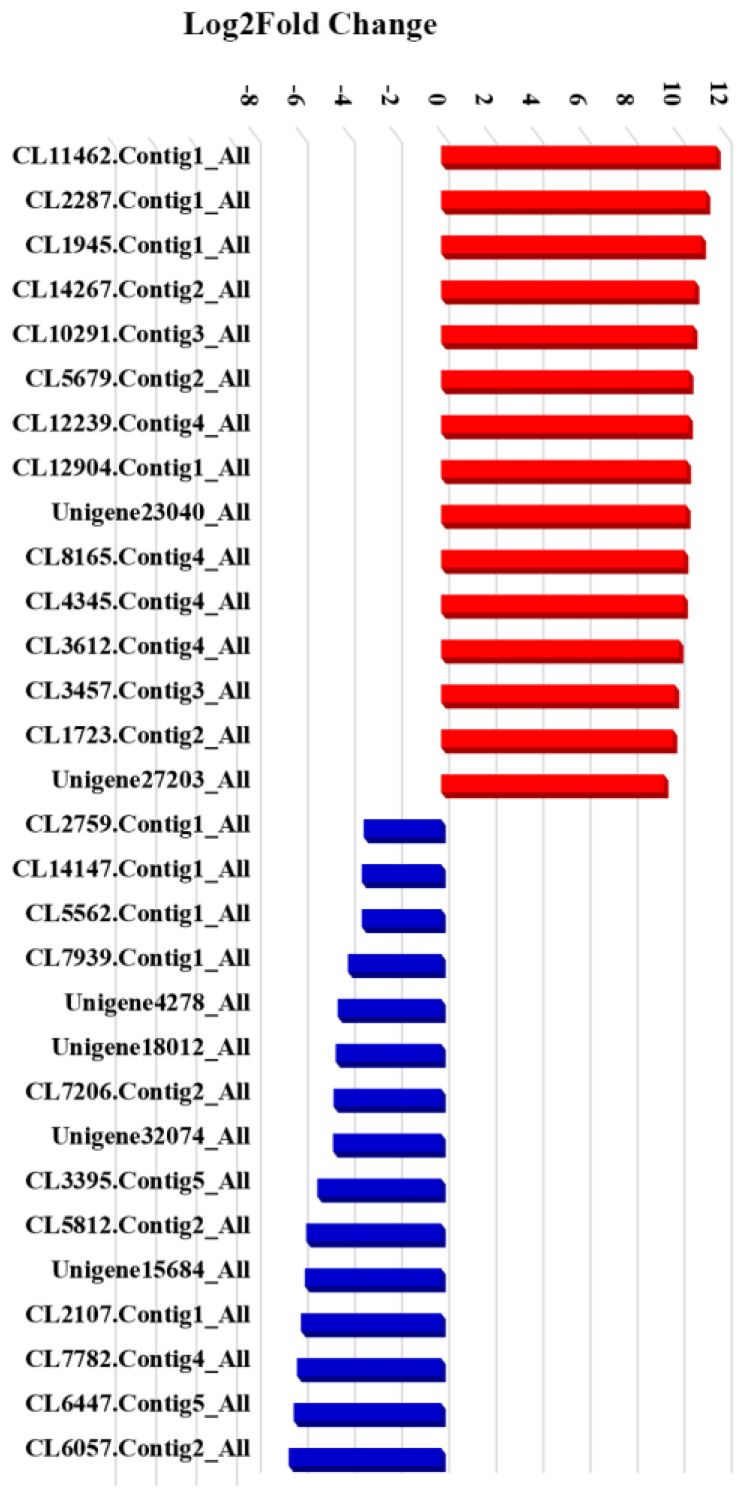
Log2-fold change results for differentially expressed unigenes involved in the rhodomyrtone content of the low- and high-rhodomyrtone genotypes of *R. tomentosa*. The red and blue colors represent the up- and downregulated genes, respectively.

**Figure 16 plants-12-03156-f016:**
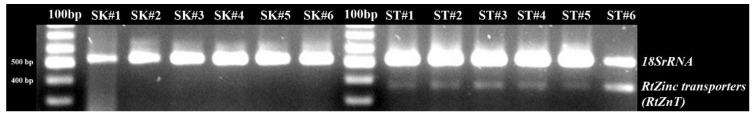
Analysis of zinc transporter (*RtZnT*) expression in relation to rhodomyrtone content in low- and high-rhodomyrtone samples (Songkhla (SK1-6) and Surat Thani genotypes (ST1-6), respectively).

**Figure 17 plants-12-03156-f017:**
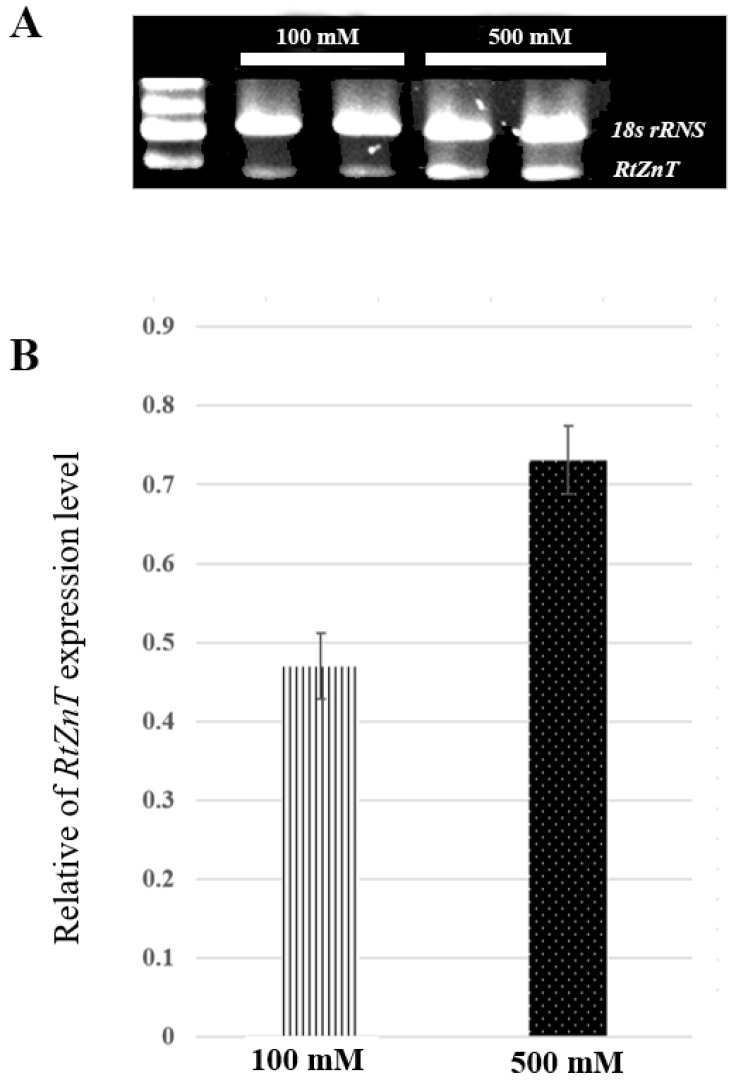
Effect of ZnSO4 on the expression profiles of *RtZnT* in *R. tomentosa* grown on 100 mM and 500 mM ZnSO4 for 7 days. (**A**) Analysis of *RtZnT* expression upon ZnSO_4_ treatment monitored by RT-PCR and using *18srRNA* as an internal control. (**B**) Analysis of the expression level of *RtZnT* on 100 mM and 500 mM ZnSO_4_. Values represent the averaged expression data for two biological replicates. Error bars indicate standard errors (SE) at *p* < 0.01.

**Table 1 plants-12-03156-t001:** Quality metrics of clean reads and transcripts from *R. tomentosa*.

Sample	RtSK1	RtSK2	RtST1	RtST2
Quality of clean reads	Distribution of base quality on clean reads			
Total raw reads (M)	75.37	77.13	78.86	30.63
Total clean reads (M)	65.68	66.48	66.03	65.34
Total clean bases (Gb)	9.85	9.97	9.90	9.80
Clean reads Q20 (%)	96.46	95.32	95.16	94.95
Clean reads Q30 (%)	91.63	86.53	86.19	86.73
Clean reads ratio (%)	87.13	86.20	83.71	81.04
Quality metrics of transcripts from de novo assembly				
Total number	157,501	118,921	103,157	170,364
Total length	81,844,035	63,411,497	58,750,215	81,846,852
Mean length	519	533	569	480
N50	869	934	1034	699
N70	363	395	461	325
N90	216	216	219	212
GC (%)	48.18	48.64	47.75	47.31

**Table 2 plants-12-03156-t002:** Quality metrics of *R. tomentosa* unigenes.

Sample	Total Number	Total Length	Mean Length	N50	N70	N90	GC (%)
RtSK1	96,036	62,329,632	649	1158	533	249	47.85
RtSK2	74,548	49,340,615	661	1168	594	247	48.23
RtST1	66,782	46,635,444	698	1244	659	254	47.75
RtST2	98,941	60,643,479	612	1050	462	246	47.31
All unigenes	186,033	108,705,360	584	1049	420	232	48.18

**Table 3 plants-12-03156-t003:** Summary of the annotation of *R. tomentosa* unigenes.

Values	Total	Nr	Nt	Swiss-Prot	KEGG	KOG	InterPro	GO	Intersection	Overall
Number	186,033	83,402	58,242	54,573	63,831	68,929	74,102	48,439	18,654	103,616
Percentage	100%	44.83%	31.31%	29.34%	34.31%	37.05%	39.83%	26.04%	10.03%	55.70%

**Table 4 plants-12-03156-t004:** Detection and expression of unigenes involved in rhodomyrtone content of the low- and high-rhodomyrtone *R. tomentosa* genotypes.

GeneID	Log2Fold Change	Up/Down-Regulation	Nr Predicted
CL11462.Contig1_All	11.68	Up	PREDICTED: ethylene-responsive TF ERF014
CL2287.Contig1_All	11.24	Up	PREDICTED: umecyanin isoform X1
CL1945.Contig1_All	11.06	Up	PREDICTED: zinc transporter
CL14267.Contig2_All	10.77	Up	PREDICTED: myb-like protein H
CL10291.Contig3_All	10.70	Up	PREDICTED: ACR1 protein
CL5679.Contig2_All	10.52	Up	PREDICTED: ABR1transcription factor
CL12239.Contig4_All	10.50	Up	PREDICTED: ubiquinol oxidase 3
CL12904.Contig1_All	10.42	Up	PREDICTED: WRKY 42
Unigene23040_All	10.40	Up	PREDICTED: expansin-like B1
CL8165.Contig4_All	10.31	Up	PREDICTED: alpha-amylase
CL4345.Contig4_All	10.29	Up	PREDICTED: cathepsin B
CL3612.Contig4_All	10.10	Up	14-3-3 protein 4
CL3457.Contig3_All	9.92	Up	PREDICTED: laccase-15
CL1723.Contig2_All	9.84	Up	PREDICTED: mannitol dehydrogenase
Unigene27203_All	9.46	Up	PREDICTED: polyphenol oxidase
CL2759.Contig1_All	−3.29	Down	PREDICTED: serine-glyoxylateamino transferase
CL14147.Contig1_All	−3.37	Down	PREDICTED: oxygen-evolving enhancer protein 1
CL5562.Contig1_All	−3.37	Down	PREDICTED: oxygen-evolving enhancer protein 2
CL7939.Contig1_All	−3.96	Down	PREDICTED: MgPME
Unigene4278_All	−4.39	Down	PREDICTED: granule-bound starch synthase 1
Unigene18012_All	−4.48	Down	PREDICTED: plastocyanin
CL7206.Contig2_All	−4.57	Down	PREDICTED: ferredoxin-NADP reductase
Unigene32074_All	−4.59	Down	PREDICTED: basic proline-rich protein-like isoform X1
CL3395.Contig5_All	−5.26	Down	PREDICTED: GAPDH
CL5812.Contig2_All	−5.73	Down	PREDICTED: rbcl/oxygenase small subunit
Unigene15684_All	−5.79	Down	PREDICTED: photosystem I reaction center subunit N
CL2107.Contig1_All	−5.95	Down	PREDICTED: catalase isozyme 1
CL7782.Contig4_All	−6.12	Down	PREDICTED: agamous-like MADS-box protein, AGL65
CL6447.Contig5_All	−6.26	Down	PREDICTED: peroxisomal (S)-2-hydroxy-acid oxidase
CL6057.Contig2_All	−6.47	Down	PREDICTED: chlorophyll a-b binding protein

## Data Availability

The original contributions presented in the study are included in the article/Appendix A. Further inquiries can be directed to the corresponding author.

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
