# Peer review of "Genomic and Transcriptional Profiling Analysis and Insights into Rhodomyrtone Yield in Rhodomyrtus tomentosa (Aiton) Hassk"

_plants, 2023, doi:10.3390/plants12173156_

Round 1

Reviewer 1 Report

The manuscript "Genomic and Transcriptional Profiling Analysis and Insights into Rhodomyrtone Yield in Rhodomyrtus tomentosa (Aiton) Hassk " is interesting and a good basis for achieving a high yield of Rhodomyrtone in R. tomentosa. The article is well structured, and adequate analyzes have been selected to achieve the desired results. My main comments are on the way the results are presented.

Some remarks:

Unify the spelling of the name of the species you are working with: R. tomentosa; R. tomentora; R. Tomentora - must be in italic!

Lines 129 - 130: This sentence is redundant!

Line 137: "blasts" must be with upper cases!

Figure 1C will be better to be Figure 2.

Figure 1C: on the y-axis is written "Quantitation analysis", but below the figure is written "Qaulity analysis"?!

Figure 1C does not write what is plotted on the abscissa and ordinate and what the unit of measurement is! Also, in the Figure on both bars, there are deviations that are not marked what they show!

Figure 2 does not write what is plotted on the abscissa and ordinate and what the unit of measurement is!

In the section: Results and Discussion - 2.5 and 2.6 must be combined as one subsection!

Figure 11: I think one of them - Figure 11A or Figure 11B is enough!

Figure 12A: “GO classification of DEGs that represents number of DEG and axis represents GO term.” Which axis? Why is Figure 12A separated by a paragraph from Figure 12B? Why not Figure 12 and Figure 13?

Table 4: Extremely difficult to read and understand! Find an optimal way of presenting this data!

Line 379: "RT–qPCR Validation" is qRT-PCR.

Figure 15 must be A and B. In Figure 15 (first part), it is not written what is #1 and #2 on the gel!

Line 496: "In this experiment, four DEGs were randomly selected." Lines 380 - 381: "To check the accuracy of the RNA-seq results, three genes associated with Rhodomyrtone were randomly selected for semi-qRT-PCR validation." Finally, the authors validated just one gene by semi-qRT-PCR! Explain, please!

The origin of the primers for DNA barcoding needs to be clarified: ITS and matK. The authors design them, or they are from an article! I saw the sentence on line 498: "The Table provides information on the design of the semi qRT–PCR primers." Which Table?

matK sometimes is in italic; sometimes it is not!

The authors performed 40 amplification cycles for semi-quantitative RT-PCR, but this condition is saturating. Usually, up to 26 - 27 cycles are applied.

In the Conclusions section, between lines 519 - 522: "In addition, semi-qRT-PCR analysis revealed that zinc transporter protein (CL1945.Contig1_All; RtZnT) was highest in DEG analysis and was highest in the Surat Thani genotype, which had high Rhodomyrtone content." Using this analysis, the authors confirmed or validated the result from transcriptome data but did not reveal it!

Author Response

Dear reviewer,

Thank you very much for your valuable comments.

All comments and suggestions will be answered as follows and Please see the attachment.

Best regards,

Alisa Nakkaew 

Some remarks:

Unify the spelling of the name of the species you are working with: R. tomentosa; R. tomentora; R. Tomentora - must be in italic!

Response: corrected all to “R. tomentosa

Lines 129 - 130: This sentence is redundant!

Response:  deleted Lines 129 – 130 and corrected sentence as Lines 130-131

Line 137: "blasts" must be with upper cases!

Response: corrected to “using BLASTN” line 128

Figure 1C will be better to be Figure 2.

Response: corrected all to “Figure 2”

Figure 1C: on the y-axis is written "Quantitation analysis", but below the figure is written "Qaulity analysis"?!

Response: corrected to “Quantitation analysis”

Figure 1C does not write what is plotted on the abscissa and ordinate and what the unit of measurement is! Also, in the Figure on both bars, there are deviations that are not marked what they show!

Response: corrected “Figure 2”

Figure 2 does not write what is plotted on the abscissa and ordinate and what the unit of measurement is!

Response: corrected “Figure 2”

In the section: Results and Discussion - 2.5 and 2.6 must be combined as one subsection!

Response:   combined section: 2.5 and 2.6 as one subsection line 294-323

Figure 11: I think one of them - Figure 11A or Figure 11B is enough!

Response: corrected to “Figure 11. Scatter are representing log10 transformed…...”

Figure 12A: “GO classification of DEGs that represents number of DEG and axis represents GO term.” Which axis? Why is Figure 12A separated by a paragraph from Figure 12B? Why not Figure 12 and Figure 13?

Response: corrected “Figure ” as line 332-338

Table 4: Extremely difficult to read and understand! Find an optimal way of presenting this data!

Response: corrected “Table 4 ” line 378-384

Line 379: "RT–qPCR Validation" is qRT-PCR.

Response: corrected to “qRT-PCR”

Figure 15 must be A and B. In Figure 15 (first part), it is not written what is #1 and #2 on the gel!

Response: corrected “Figure 15”

Line 496: "In this experiment, four DEGs were randomly selected."

Lines 380 - 381: "To check the accuracy of the RNA-seq results, three genes associated with Rhodomyrtone were randomly selected for semi-qRT-PCR validation.

" Finally, the authors validated just one gene by semi-qRT-PCR! Explain, please!

Response: corrected sentence in section 2.7 as “Line 397-391”

Initially, three DEGs (RtZnT, Rtlaccase15, and RtERF14) were randomly selected and the accuracy of the RNA-seq results was subsequently confirmed. However, only RtZnT was further amplified as it was which is associated with rhodomyrtone content.

The origin of the primers for DNA barcoding needs to be clarified: ITS and matK. The authors design them, or they are from an article!

Response: corrected “the origin of the primers for DNA barcoding information “Line 441-445

I saw the sentence on line 498: "The Table provides information on the design of the semi qRT–PCR primers." Which Table?

Response: corrected sentence” in section 3.6 as “Line 512”and add “Supplementary Table 1

matK sometimes is in italic; sometimes it is not!

Response: corrected all to “matK

The authors performed 40 amplification cycles for semi-quantitative RT-PCR, but this condition is saturating. Usually, up to 26 - 27 cycles are applied.

Response: For a normal semiquantitative RT-PCR, up to 25-30 amplification cycles were performed, but we found that the PCR products of RtZnT were low, so we decided to use 40 amplifications because we wanted to show the clear differential expression of genes between low and high rhodomyrton genotypes.

In the Conclusions section, between lines 519 - 522: "In addition, semi-qRT-PCR analysis revealed that zinc transporter protein (CL1945.Contig1_All; RtZnT) was highest in DEG analysis and was highest in the Surat Thani genotype, which had high Rhodomyrtone content." Using this analysis, the authors confirmed or validated the result from transcriptome data but did not reveal it!

Response: corrected sentence” in In the Conclusions section as “Line 530-534”

Reviewer 2 Report

L34-L41. The description of this part may deletion of shorten. we perform research because Aiton is valuable and economic importance, thus authors may focus on its industry, output value, planting area (better using statistical data to support).

L55, L58. ml or mL. Thoroughly check the manuscript.

L68. May change however to meanwhile

L70. Reference for Yao is missing.

L76. L337. “Rhodomyrtone s”?

L90. The sentence “it can be a successful tool for marker-based” is uncompleted.

L114. Italic de novo. Thoroughly check the manuscript.

L127. Italic the Latin name. Thoroughly check the manuscript.

L137. Change blasts to blastn.

L153. “accurately This”?

L173. It seems the Y-axis missing unit.

Table 1. The blank missing before some (.

L248. Missing thousand separator for some number.

Figure 12A. The right bar is no longer enough to cover the range.

It is strange for Figure 12A and 12B in two figures. Usually, A and B are merged in one figure and indicated by (A) and (B).

Major point. The language should be critical edited.

The language should be critical edited.

Author Response

Dear reviewer,

Thank you very much for your valuable comments.

All comments and suggestions will be answered as follows and Please see the attachment.

Comments and Suggestions for Authors

L34-L41. The description of this part may deletion of shorten. we perform research because Aiton is valuable and economic importance, thus authors may focus on its industry, output value, planting area (better using statistical data to support).

Response:  deleted L34-L41

L55, L58. ml or mL. Thoroughly check the manuscript.

Response: corrected all to “mL”

L68. May change however to meanwhile

Response: corrected to “Meanwhile”

L70. Reference for Yao is missing.

Response: corrected Reference for Yao [9] as L560-561

  1. Yao, X., Mating System and Genetic Diversity of Rhodomyrtus tomentosa (Myrtaceae) Detected by ISSR Markers, in Master’s Thesis. University of Hong Kong; Pokfulam, Hong Kong. 2010.

L76. L337. “Rhodomyrtone s”?

Response: corrected to “Rhodomyrtone”

L90. The sentence “it can be a successful tool for marker-based” is uncompleted.

Response: corrected to “it can be a successful tool for marker-based selection in medicinal plants” L99

L114. Italic de novo. Thoroughly check the manuscript.

Response: corrected all to “de novo

L127. Italic the Latin name. Thoroughly check the manuscript.

Response: corrected all manuscript

L137. Change blasts to blastn.

Response: corrected all to “BLASTN”

L153. “accurately This”?

Response: corrected all to “accurately in different producing areas.”

L173. It seems the Y-axis missing unit.

Response: L173 corrected to “Quantitation analysis of Rhodomyrtone (µg/mL)”

Table 1. The blank missing before some (.

Response: corrected “Table 1”

 L248. Missing thousand separators for some number.

Response: corrected all between L248-L256

Figure 12A. The right bar is no longer enough to cover the range.

Response:  corrected “Figure 12” L340-L346

It is strange for Figure 12A and 12B in two figures. Usually, A and B are merged in one figure and indicated by (A) and (B).

Response:  corrected “Figure 12” L340-L346

Major point. The language should be critical edited.

Comments on the Quality of English Language

The language should be critical edited.

Response: corrected English Language editing by MDPI.

Round 2

Reviewer 1 Report

I take the authors' corrections and suggest that the editor accept the manuscript in its present form!

Reviewer 2 Report

Nice work for the revision.